# Association of Asymmetric Dimethylarginine and Nitric Oxide with Cardiovascular Risk in Patients with End-Stage Liver Disease

**DOI:** 10.3390/medicina56110622

**Published:** 2020-11-18

**Authors:** Maro Dragičević, Iva Košuta, Egon Kruezi, Marijana Vučić Lovrenčić, Anna Mrzljak

**Affiliations:** 1Department of Cardiology, Merkur University Hospital, 10000 Zagreb, Croatia; maro_drag@yahoo.com; 2Department of Internal Medicine, University Hospital Centre, 10000 Zagreb, Croatia; ivakosuta@gmail.com; 3Department of Gynaecology and Obstetrics, Sisters of Charity University Hospital Centre, 10000 Zagreb, Croatia; egon.kruezi@gmail.com; 4Department of Clinical Chemistry and Laboratory Medicie, Merkur University Hospital, 10000 Zagreb, Croatia; 5Department of Gastroenterology, Merkur University Hospital, 10000 Zagreb, Croatia; anna.mrzljak@mef.hr; 6School of Medicine, University of Zagreb, 10000 Zagreb, Croatia

**Keywords:** end-stage liver disease, transplantation, endothelial dysfunction, cardiovascular risk, asymmetric dimethylarginine, nitric oxide

## Abstract

*Background and objectives*: Endothelial dysfunction has been proposed to be an underlying mechanism of the pronounced cardiovascular morbidity in end-stage liver disease (ESLD), but clinical evidence is still limited. In this study, we investigated the association of circulating levels of asymmetric dimethylarginine (ADMA) and nitric oxide (NO) with estimated cardiovascular risk in patients with ESLD awaiting liver transplantation. *Materials and Methods*: ADMA and NO levels were measured in the sera of 160 adult ESLD patients. The severity of hepatic dysfunction was assessed by the model for end-stage liver disease (MELD) score. The cardiovascular risk was estimated with the European Society of Cardiology Systematic Coronary Risk Estimation (SCORE) index, which was used to dichotomize patients in the subgroups depicting higher and lower cardiovascular risk. *Results*: Severe hepatic dysfunction (MELD ≥ 18) was present in 38% of the patients, and a higher cardiovascular risk was present in almost half of the patients (*N* = 74). ADMA and NO both significantly increased with the progression of liver disease and were independently associated with higher cardiovascular risk. Fasting glucose also independently predicted a higher cardiovascular risk, while HDL cholesterol and the absence of concomitant hepatocellular carcinoma were protective factors. *Conclusions*: These results suggest a remarkable contribution of the deranged arginine/NO pathway to cardiovascular risk in patients with end-stage liver disease.

## 1. Introduction

Liver transplantation (LT) is an established treatment of choice for end-stage liver disease (ESLD), acute liver failure, and hepatocellular carcinoma (HCC) [1]. LT candidates are nowadays older and have more comorbidities. Among them, patients with traditional cardiovascular (CV) risk factors have poorer post-LT outcomes and reduced survival rate [2]. Estimating CV risk is an essential step in the evaluation of potential LT recipients, as well as during the follow-up in the post-transplantation period. However, it has been shown that CV risk assessment in LT candidates cannot sufficiently be carried out solely by traditional non-invasive tests [3].

Endothelial dysfunction (ED), which is caused by the deranged hepatic metabolism of the potent vasodilatory mediator, i.e., nitric oxide (NO), has been proposed to be a plausible underlying mechanism of CV morbidities associated with ESLD [4]. ED results from a misbalance of vasoconstrictive and vasodilatory mediators, which leads to endothelial inflammation and myofibroblast migration, causing atherosclerotic vascular disease [5]. Apart from eliciting vasodilation, NO inhibits growth and contraction of smooth muscle cells, thrombocyte aggregation, and leukocyte-endothelial adhesion [6]. NO is synthesized from L-arginine by various nitric oxide synthase (NOS) enzymes [7]. The activity of endothelial isoform (eNOS) is regulated by asymmetric dimethylarginine (ADMA), which is a biogenetic L-arginine analog derived by proteolysis of methylated proteins. ADMA inhibits the activity of eNOS via paracrine regulation [8].

Numerous studies have identified increased circulating ADMA levels as an independent CV risk factor in diverse risk groups [9,10]. In patients with ESLD, circulating ADMA levels correlated with the severity of portal hypertension [11], whereas decreased production of NO increased vascular resistance and portal hypertension [12]. It has been proposed that elevated levels of ADMA played a role in the pathogenesis of major complications of ESLD, i.e., hepatorenal syndrome [13] and hepatic encephalopathy [14], by blocking NO synthesis at the level of renal vessels and modulating an array of NO-dependent cerebral processes. Clinical interventions aimed at mitigating ESLD were able to normalize serum ADMA levels in patients treated with transjugular intrahepatic portosystemic shunt [15]. However, controversial reports exist regarding the effect of liver transplantation with both decreased [16] and increased [17] ADMA levels observed shortly after LT.

NO has a dual role in liver pathophysiology, depending on the type of the NOS catalyzing its production from L-arginine. The endothelial NOS, located in the liver sinusoidal endothelial cells, produces a small amount of NO, which is responsible for the maintenance of intrahepatic vascular tone and blood flow, as well as immune response via hepatic stellate cells and Kupfer cells. On the other hand, inflammatory mediators such as cytokines and bacterial endotoxins upregulate inducible NOS (iNOS) which is responsible for the production of a large amount of NO, able to elicit an array of damaging cellular processes ranging from apoptosis to DNA damage either per se, or by producing extremely harmful free radicals from coupling with reactive oxygen species [18].

In this study, we aimed at investigating the association of ADMA and NO with estimated CV risk in patients with ESLD awaiting liver transplantation.

## 2. Materials and Methods

### 2.1. Study Population

The study took place between April 2014 and April 2017 at the Merkur University Hospital, Zagreb, Croatia. Patients with clinically and histologically confirmed cirrhosis eligible for LT were included in the study. The exclusion criteria were the following: acute liver failure; previous solid-organ recipients; multiorgan transplantation candidates; and comorbid chronic kidney disease (CKD) defined as estimated glomerular filtration rate (eGFR) < 60 mL/min/1.73 m^2^, or histological evidence of kidney disease, or both for longer than six months.

The study population included 160 patients (114 males) with ESLD, predominantly alcohol related (52%), followed by viral (23%), cholestatic (10%), and other (15%) etiologies. Among them, 45 patients (28%) had hepatocellular carcinoma (HCC), 19 patients (12%) had diabetes, and 18 patients (11%) had previous cardiovascular disease (CVD).

The study was conducted according to the principles expressed in the Declaration of Helsinki. The eligible patients were informed about the study, and those willing to participate gave their written consent. The study was approved by the Merkur University Hospital’s Ethics Committee (03/1-1365).

### 2.2. Methods

Blood pressure was measured in a seated position using a sphygmomanometer with an appropriately sized blood pressure cuff. Hypertension was defined as systolic blood pressure values > 140 mmHg and/or diastolic blood pressure values > 90 mmHg, as defined by ESC/ESH guidelines [19]. Body weight (BW, kg) and body height (BH, m) was measured using a balance beam scale with light clothing without shoes and a wall-mounted stadiometer, respectively. Body mass index (BMI, kg/m^2^) was calculated according to formula BW/(BH).

Total CV risk was estimated with the European Society of Cardiology 2012 SCORE (Systematic Coronary Risk Estimation), an integrated surrogate index calculated using online interactive tool recalibrated for use in the Croatian population, available at http://www.heartscore.org/hr_HR. Calculation is based on gender, age, smoking, systolic blood pressure, and total cholesterol and estimates the risk of fatal CV disease events over ten years [20]. The subjects with documented CVD, comorbid diabetes with target organ damage, or with a significant risk factor (smoking, hypertension, diabetes, or dyslipidemia), and those with severe CKD (eGFR ≤ 30 mL/min/1.73 m^2^) were classified within the highest CV risk category regardless of the calculated score. The results were classified into the recommended four CV-risk categories, i.e., very high risk, high risk, moderate risk, and low risk, respectively, and further dichotomized into cumulative ranks depicting higher (very high + high risk) and lower (moderate + low risk) CV risk classes for this study.

Phlebotomy was performed at fasting, using appropriate vacuum tubes (Becton Dickinson, Franklin Lakes, NY, USA). The blood samples were processed by centrifugation (3000× *g*, 10 min) and separation of plasma and serum samples for further testing. Fresh samples were used for routine laboratory testing, while aliquots of sera were stored (−70 °C) immediately after separation from cells for the analysis of ADMA and NO. Lipid profile (total, HDL, and LDL cholesterol, as well as triglycerides), bilirubin and glucose were measured using routine laboratory methods on an automated analytical platform (AU680, Beckman Coulter, Brea, CA, USA). Serum creatinine was measured by an enzymatic (Beckman Coulter, Inc., Pasadena, CA, United States) assay with an intra-assay imprecision (CV) of 1.39%, and GFR estimated by 4-variable CKD-EPI equation. The plasma international normalized ratio (INR) was derived from prothrombin time results determined using an automated coagulation analyzer (Sysmex 2100i, Siemens Healthineers, Erlangen, Germany). Fasting insulin was measured using an automated chemiluminescence immunoassay (Advia Centaur XP, Siemens Healthineers, Malvern, PA, USA). Homeostasis model assessment HOMA2 calculator (version 2.2.2, Diabetes Trials Unit, University of Oxford, available at http://www.dtu.ox.ac.uk/homacalculator/index.php, was used to estimate beta cell function (HOMA2-B (%)) and insulin sensitivity (HOMA2-IS (%)) from fasting glucose and insulin concentrations [21].

The model for end-stage liver disease score (MELD), a clinically established index of the severity of hepatic dysfunction was calculated according to the following formula: MELDScore = 10 × ((0.957 × ln(Creatinine)) + (0.378 × ln(Bilirubin)) + (1.12 × ln(INR))) + 6.43 available at: https://reference.medscape.com/calculator/meld-score-end-stage-liver-disease [22]. An increasing MELD score is associated with increasing severity of hepatic dysfunction and increasing the three-month mortality risk in patients with ESLD. For depicting the severity of hepatic dysfunction, two subgroups were derived at a clinically meaningful MELD cut-off set at the MELD value of 18. This arbitrary cut-off was selected due to the specific circumstances of our transplant community, i.e., high donation rate and extremely short waiting time for liver transplantation.

ADMA levels were determined by the competitive enzyme-linked immunosorbent assay (ELISA) with declared expected values ranging from 0.40 to 0.85 μmol/L and a sensitivity of 0.05 μmol/L (DLD Diagnostics, GmbH, Hamburg, Germany), as previously described [23]. Due to a very short half-life, NO was determined, indirectly, as the sum of total nitrite and nitrate concentration measured by a commercially available method based on Griess reaction (Total Nitric Oxide and Nitrate/Nitrite Parameter Assay Kit, R&D Systems, Minneapolis, MN, USA). The levels of nitrite and nitrate were determined in serum samples cleared from proteins by ultrafiltration using 10,000 molecular weight cut-off filters (Amicon Ultra 0.5, Millipore, Burlington, MA, USA).

### 2.3. Statistical Analysis

After testing for normality (Kolmogorov–Smirnov test), continuous data were expressed as median and interquartile range (IQR). Categorical variables are given as numbers (n) and proportions (%) of patients with a specific characteristic. The comparisons between groups were tested with either Student’s *t*-test or Mann–Whitney U test, depending on the data distribution. Data expressed as the proportion were tested with Pearson’s χ^2^ test. Correlations were assessed by Spearman’s rank test. A binary logistic regression model was built to evaluate independent risk factors for elevated CV risk in our patients. Statistical analyses were carried out using MedCalc Statistical Software version 18.11.6 (MedCalc Software bvba, Ostend, Belgium) and Statistica Software version 13.3 (Tibco Software, Inc., Palo Alto, CA, USA). A value of *P* < 0.05 was considered to be statistically significant.

## 3. Results

### 3.1. Etiology and Severity of ESLD

The severity of hepatic dysfunction was classified according to the MELD cut-off set at 18, i.e., MELD < 18 and MELD ≥ 18. Clinical and laboratory characteristics of the study population are presented in Table 1. The patients in the MELD ≥ 18 group had higher ADMA, NO, and CRP levels, and lower total, HDL, and LDL cholesterol, and triglycerides than those in MELD < 18 group. Fasting plasma glucose, insulin, HOMA2-B, and HOMA2-IS, as well as the occurrence of previous cardiovascular morbidity and diabetes did not differ between the groups. However, hypertension and HCC were more prevalent in the MELD < 18 group. The SCORE-based CV risk categories were evenly distributed between both groups (χ^2^ test, *p* = 0.391).

ADMA and NO levels did not differ among subjects according to the etiology of liver disease (alcohol related vs. viral vs. cholestatic vs. other, respectively, *P* = 0.599 and *P* = 0.875, Kruskal–Wallis test, not shown).

### 3.2. Cardiovascular Risk

Patients with ESLD were classified based on the SCORE index assessment in two categories depicting higher (N = 73) and lower (N = 87) CV risk. The higher CV risk group had significantly higher ADMA and NO (Figure 1), as well as FPG and CRP levels, and lower HDL and LDL cholesterol levels (Table 2). No differences were observed in other clinical and laboratory parameters (Table 2). There was also no difference regarding the etiology of liver disease between the patients with higher and lower CV risk (χ^2^ test, *P* = 0.919).

### 3.3. Associations between Variables

Rank correlation analysis showed significant (*P* < 0.05) weak to moderate positive correlations among ADMA, MELD, and HOMA2-IS (Spearman’s ρ = 0.161 and 0.197) and between NO and MELD (Spearman’s ρ = 0.331). Significant negative correlations (*P* < 0.05) were also observed between ADMA and total, as well as LDL cholesterol (Spearman’s ρ = −0.179 and −0.195), and between NO and total, HDL, and LDL cholesterol (Spearman’s ρ = −0.326, −0,273, and −0.269, respectively) (not shown).

Stepwise logistic regression analysis was carried out to assess the predictors of CV risk in our patients. The model included SCORE index-based CV risk categories as binary dependent variable; ADMA, NO, CRP, FPG, MELD score, HDL cholesterol, HOMA2-B, and HOMA2-IS as continuous independent predictors; and presence/absence of HCC as a categorical factor (Table 3). The model was statistically significant (χ2 (5) = 27.4, *P* < 0.001) and was able to explain 24.6% of the variance (Nagelkerke R2). FPG, ADMA, NO, and HDL cholesterol levels were identified as significant predictors of CV risk, with HCC as a biological confounder. The model correctly classified 68% of the cases and provided the area under the ROC curve 0.737.

## 4. Discussion

In this study, we investigated the association of two established biomarkers of endothelial dysfunction, i.e., ADMA and NO, with cardiovascular risk in patients with ESLD awaiting liver transplantation. Our results show that both ADMA and NO (a) significantly increase with the progression of liver disease, and (b) are independently associated with elevated CV risk. A higher fasting plasma glucose was also identified as a significant predictor, while HDL cholesterol levels and the absence of hepatocellular carcinoma were protective factors of CV risk. Almost half of our patients (46%) had elevated CV risk.

Berzigotti et al. [24] reported on the high frequency of CV risk and vascular endothelial dysfunction, as measured with ultrasound-based flow-mediated dilatation, in patients with cirrhosis. The results of our study are in accord with their findings, but also reveal, for the first time, that elevated CV risk in patients with cirrhosis can be predicted with biomarkers of endothelial dysfunction, ADMA and NO. In a systematic review and meta-analysis of 22 prospective studies, Willeit et al. reported that elevated levels of ADMA were consistently associated with an increased CV risk both in general and diverse clinical populations, including those with pre-existing CVD and kidney disease [25]. On the basis of ample experimental evidence, a reduced bioavailability of NO due to the inhibitory action of ADMA was proposed to play a pivotal role in the development of endothelial dysfunction, a significant early event in the pathogenesis of cardiovascular diseases [4].

The liver plays a crucial role in ADMA elimination through enzymatic degradation catalyzed by an isoform of an enzyme dimethylarginine dimethylaminohydrolase (DDAH I), which is most abundantly expressed in hepatocytes. Experimental evidence has indicated that bile duct ligation, in an established rodent model of liver cirrhosis, resulted in elevated ADMA levels [26]. In humans, increased plasma levels of ADMA were found in alcoholic liver disease [27], non-alcoholic fatty liver disease (NAFLD) [28,29], and chronic hepatitis B [30]. However, unchanged plasma levels of ADMA were reported in chronic hepatitis C [31], suggesting that methylarginine metabolism in conditions with severe hepatocellular damage may be affected beyond a decreased clearance of ADMA. Indeed, increased generation of ADMA may also result as a consequence of the upregulation of the key enzyme (protein arginine N-methyltransferase (PRMT)) responsible for its synthesis, as evidenced in patients with alcoholic hepatitis [32]. The results of our study not only confirm previous reports on the increasing ADMA levels with the aggravation of end-stage liver disease but also reveal, for the first time, that higher ADMA levels independently contribute to the CV risk in patients with liver cirrhosis (Table 3). A similar effect towards CV risk was observed with NO levels that were ealso higher in patients with MELD ≥ 18 (Table 1). Considering ADMA’s biological role as a potent endogenous inhibitor of NOS, an increase in ADMA would be expected to elicit a decrease in NO levels. However, this was not the case in our study. NO is one of the essential molecular effectors released from cirrhotic liver and is considered to be responsible for vasodilating and cardio-suppressive action leading to circulatory dysfunction in advanced liver disease [33]. Activation of the inducible isoform of NOS seems to be responsible for this phenomenon. Ample experimental evidence supports a central role of oxidative and nitrosative stress in the pathogenesis of NAFLD, alcoholic liver disease, HCC, and viral hepatitis, due to pronounced induction of iNOS by diverse molecular mechanisms involved in these diseases [25,34]. Clinical studies conducted so far have revealed an increase in total NO in ESLD, a positive correlation with markers of inflammation and clinical signs of portal hypertension [35], and an increased release of NO in the splanchnic circulation despite decreased activity of eNOS in the cirrhotic liver [36]. However, no difference in NO levels has been found between compensated and decompensated cirrhosis by some [37], but not all authors [27], as well as in patients with chronic hepatitis C [31], indicating, as was the case with ADMA, the existence of very complex and diverse mechanisms affecting the arginine/NO pathway in various liver diseases. Our study revealed no significant differences in ADMA and NO levels between different liver disease etiologies but having in mind the specific cohort of liver transplant candidates with ESLD, these results should be interpreted with caution.

Evidence collected, so far, regarding the association of deranged NO metabolism with cardiovascular pathology in chronic liver disease has been scarce. In 2000, Liu et al. found that experimentally induced NO overproduction via stimulation of iNOS elicited a decrease in cardiac contractility in rats [38], indicating a significant role for iNOS in the pathogenesis of cirrhotic cardiomyopathy. This mechanism offers a plausible explanation for our results which showed that elevated ADMA levels were not able to reduce NO in cirrhotic patients. Most likely, overproduction of NO was a result of the activation of iNOS modulated by inflammation, that could not be inhibited by ADMA. The involvement of inflammation-mediated upregulation of NO production is supported by our finding of the significantly elevated CRP in patients with severe hepatic dysfunction (MELD > 18, Table 1) and with the higher CV risk (Table 2). However, the intensity of inflammation, as measured with CRP, was not directly associated with elevated CV risks in our patients, nor was insulin resistance, which is another pathophysiological mechanism closely related to both endothelial [39] and liver dysfunction [40]. Although our patients with an average HOMA2-IS of 40% were insulin resistant, this feature was not influenced by either the severity of liver dysfunction or the magnitude of total CV risk. This finding may also support the activation of iNOS as the most probable source of elevated NO levels in our patients, since an insulin-resistant state resulted in the impairment of vascular response and increased cardiovascular risk via diverse molecular mechanisms, including inhibition of eNOS [39,40].

Nevertheless, we found that fasting plasma glucose contributed significantly to CV risk, confirming a well-known association between hyperglycemia in CV morbidity in the cirrhotic population [41]. Our finding on the protective effect of HDL cholesterol indicates that favorable lipid profile maintains its biological activity associated with CV risk in our patients, despite substantial disorders of lipid metabolism in cirrhosis [42,43]. However, the second protective factor identified in our study, the absence of malignant disease (hepatocellular carcinoma), could be related to an increasing body of evidence connecting cardiovascular diseases with cancer [44]. This interesting association should be further investigated, since our study was not designed to evaluate the association between hepatic malignancy and CV risk.

There are several limitations to our study. Our population included only patients who were eligible for liver transplantation, which may have introduced a bias in interpreting results across the broad clinical entity of ESLD. Most of the patients were male, which may have led to a selection bias, but it should be emphasized that SCORE, which was used as a measure of CV risk in this study, accounted for the variable of gender. Furthermore, we did not have a control group of healthy subjects. The CV risk estimation method used in our study, by using a score generated from age, gender, total cholesterol, and blood pressure, may not be optimal for this specific population, considering a substantial negative influence of liver dysfunction on the lipid metabolism and blood pressure. Indeed, our study confirmed previous reports on the significantly lower levels of total, HDL, and LDL cholesterol, as well as triglycerides in the more advanced stages of liver dysfunction [43], while the proportion of the patients with hypertension was significantly lower in patients with MELD ≥ 18 (Table 1). The use of clinically relevant endpoints, such as cardiovascular mortality and major adverse cardiovascular events was not possible due to the cross-sectional study design. Finally, creatinine-based eGFR could have biased the kidney function assessment with possible implications regarding ADMA levels. However, despite these limitations, our study revealed the progressive increase in, and significant contribution of, ADMA and NO levels associated with CV risk in advanced hepatic dysfunction of liver cirrhosis.

## 5. Conclusions

This study demonstrated, for the first time, an independent association of ADMA and NO with elevated CV risk in the patients awaiting liver transplantation. Apart from the endothelial dysfunction biomarkers, fasting plasma glucose was also associated with elevated CV risk, while HDL cholesterol and the absence of HCC had a protective effect in patients with ESLD. Despite the limitations, our study indicates that elevated CV risk is highly prevalent and independent of the etiology of end-stage liver disease. Instead, a cluster of endothelial and metabolic abnormalities operative in the ESLD contributed in a specific manner to the elevated CV risk in our patients. Further follow-up studies are warranted, in order to evaluate and to assess the prognostic utility of ADMA and NO as cardiovascular risk markers in the pre- and post-transplant period.

## Figures and Tables

**Figure 1 medicina-56-00622-f001:**
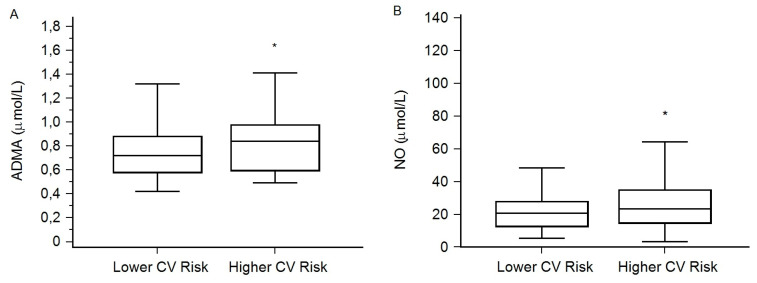
Asymmetric dimethylarginine ADMA (**A**) and total nitric oxide (NO) (**B**) concentrations between the categories of cardiovascular risk. Variables are presented as medians (the horizontal line) (interquartile range (the box)), while error bars represent minimum and maximum values. ADMA = 0.72 (0.58–0.88) vs. 0.84 (0.59–0.99) µmol/L, *P* = 0.046 and NO = 20.8 (13.0–28.0) vs. 23.9 (15.1–35.10 µmol/L, *P* = 0.044, in the groups of ESLD patients with lower and higher CV risk, respectively. * *P* < 0.05 (Mann–Whitney U test). ADMA, asymmetric dimethylarginine; NO, total nitric oxide; ESLD, end-stage liver disease.

**Table 1 medicina-56-00622-t001:** Clinical characteristics of end-stage liver disease patients, overall and by the severity of liver disease (model for end-stage liver disease (MELD)-related categories, <18 and ≥18).

Characteristics	Overall (N = 160)	MELD < 18 (N = 100)	MELD ≥ 18 (N = 60)	*P* (MELD < 18 vs. ≥ 18)
ADMA (µmol/L)	0.75 (0.59–0.94)	0.70 (0.58–0.91)	0.84 (0.64–0.97)	0.013
NO (µmol/L)	22.4 (14.2–31.8)	19.9 (13.0–27.9)	27.1 (18.7–47.3)	<0.001
Age (years)	60 (52–64)	60 (54–64)	59 (51–64)	0.971
Male (%)	71	72	68	0.623
Hypertension (%)	29	36	18	0.018
Previous CVD (%)	7	5	10	0.324
Diabetes (%)	6	6	5	0.791
HCC (%)	28	39	10	<0.001
SCORE-based CV Risk				
Low (%)	54	57	50	0.391
High (%)	46	43	50	
BMI (kg/m^2^)	25.9 (23.3–29.0)	26.3 (23.6–29.1)	25.5 (23.0–28.7)	0.388
CRP (mg/L)	7.9 (2.9–18.7)	5.2 (2.7–12.4)	11.7 (6.6–25.2)	<0.001
Total cholesterol (mmol/L)	3.35 (2.50–4.50)	3.80 (2.90–4.90)	2.60 (91.60–3.50)	<0.001
HDL cholesterol (mmol/L)	0.87 (0.60–1.20)	0.97 (0.70–1.29)	0.69 (0.35–1.00)	<0.001
LDL cholesterol (mmol/L)	2.00 (1.40–2.98)	2.35 (1.60–2.95)	1.70 (1.00–3.24)	0.010
Triglycerides (mmol/L)	0.80 (0.60–1.20)	0.91 (0.68–1.23)	0.60 (0.51–1.88)	<0.001
FPG (mmol/L)	5.4 (4.9–6.2)	5.4 (5.0–6.2)	5.4 (4.8–6.1)	0.564
Insulin (pmol/L)	117 (81–188)	123 (87–188)	108 (78–195)	0.456
HOMA2-B (%)	171 (127–234)	174 (139–231)	159 (110–245)	0.542
HOMA2-IS (%)	40 (25–58)	37 (25–55)	42 (24–60)	0.408

Data are expressed as median (interquartile range). Mann–Whitney U test between MELD-related categories. ADMA, asymmetric dimethylarginine; NO, total nitric oxide; HCC, hepatocellular carcinoma; MELD, model for end-stage liver disease score; FPG, fasting plasma glucose; HOMA2-B, homeostatic model assessment 2-derived beta cell-function; HOMA2-IS, homeostatic model assessment 2-derived insulin sensitivity.

**Table 2 medicina-56-00622-t002:** Clinical characteristics of end-stage liver disease patients by the cardiovascular-risk categories.

Characteristics	Lower CV Risk (N = 87)	Higher CV Risk (N = 73)	*P* (Lower vs. Higher CV Risk)
Male (%)	64	78	0.058
HCC (%)	23	34	0.114
MELD	16 (13–19)	17 (11–21)	0.515
BMI (kg/m^2^)	26.0 (23.1–28.8)	25.59 (23.6–29.4)	0.546
CRP (mg/L)	6.0 (2.6–13.0)	10.8 (3.6–28.9)	0.012
HDL cholesterol (mmol/L)	0.97 (0.63–1.28)	0.75 (0.52–1.04)	0.022
LDL cholesterol (mmol/L)	2.30 (1.60–3.10)	1.80 (1.06–2.70)	0.041
Triglycerides (mmol/L)	0.80 (0.63–1.22)	0.77 (0.58–1.16)	0.844
FPG (mmol/L)	5.2 (4.8–5.8)	5.7 (5.0–6.6)	0.001
Insulin (pmol/L)	114 (81–191)	123 (82–187)	0.770
HOMA2-B (%)	179 (141–240)	166 (110–219)	0.134
HOMA2-IS (%)	41 (25–61)	38 (24–55)	0.528
IRI	2.5 (1.6–4.0)	2.6 (1.8–3.9)	0.596

Data are expressed as median (interquartile range). Mann–Whitney U test between SCORE-assigned CV-risk categories. HCC, hepatocellular carcinoma; MELD, model for end-stage liver disease score; FPG, fasting plasma glucose; HOMA2-B, homeostatic model assessment 2-derived beta cell-function; HOMA2-IS, homeostatic model assessment 2-derived insulin sensitivity.

**Table 3 medicina-56-00622-t003:** Multivariate logistic regression analysis of the parameters associated with cardiovascular risk in liver transplantation candidates.

Variable	Odds Ratio	95% Confidence Interval	*P*
ADMA (µmol/L) per 1 µmol/L increase	3.6629	1.0442–12.8376	0.042
NO (µmol/L) per 1 µmol/L increase	1.0180	1.0002–1.0361	0.048
FPG (mmol/L) per 1 mmol/L increase	1.6514	1.0923–2.4966	0.017
HDL-C (mmol/L) per 1 mmol/L increase	0.4064	0.1672–0.9877	0.047
HCC absence vs. presence	0.6221	0.3921–0.9866	0.044

Variables included in the model: Dependent: SCORE-derived higher and lower CV risk categories (SCORE is estimated from gender, age, smoking, systolic blood pressure, and total cholesterol with comorbid diabetes, chronic kidney disease, and history of CVD as additional contributing variables for high CV risk scoring). Continuous independent: ADMA, asymmetric dimethylarginine; NO, total nitric oxide; CRP, C-reactive protein; FPG, fasting plasma glucose; MELD, model for end-stage liver disease score; HDL-C, HDL cholesterol; insulin; c-peptide; HOMA2-B, homeostatic model assessment 2-derived beta cell function; HOMA2-IS, homeostatic model assessment 2-derived insulin sensitivity. Categorical covariate: HCC, hepatocelullar carcinoma.

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
