# Peer review of "Association of Asymmetric Dimethylarginine and Nitric Oxide with Cardiovascular Risk in Patients with End-Stage Liver Disease"

_medicina, 2020, doi:10.3390/medicina56110622_

Round 1
Reviewer 1 Report
The current study suggesting a significant contribution of asymmetric dimethylarginine and nitric oxide to the cardiovascular risk in patients with end-stage liver disease. The study is interesting and well written. The background of the study has been clearly defined. the results are adequately discussed. I’ve only minor comment:
- This study has enrolled 160 patients with 114 males. Are there any gender differences on association of asymmetric dimethylarginine and nitric oxide with cardiovascular risk? Authors should discuss that.
- In result section, authors should move the “study population” data to the corresponding “Materials and Methods” section, where these data should be presented.
- The manuscript would benefit from additional editing for typo. For example: page 2, line 78; page 2, line 89; page 3, line 96; page 10, line 331.
Author Response
The current study suggesting a significant contribution of asymmetric dimethylarginine and nitric oxide to the cardiovascular risk in patients with end-stage liver disease. The study is interesting and well written. The background of the study has been clearly defined. the results are adequately discussed. I’ve only minor comment:
- This study has enrolled 160 patients with 114 males. Are there any gender differences on association of asymmetric dimethylarginine and nitric oxide with cardiovascular risk? Authors should discuss that.
Answer:
Thank you for this comment. Indeed, males were predominant in our study population, which reflected the general sex-distribution in our liver transplant programme.This may have led to a selection bias. However, the variable of gender was statistically controlled in our study, and we observed no differences in the biomarkers of endothelial dysfunction between males and females. We have discussed this issue by adding a sentence „Most of the patients were male, which may have led to a selection bias, but it should be emphasized that SCORE which was used as a measure of CV risk in this study accounts for the variable of gender”. (lines 354-356).
- In result section, authors should move the “study population” data to the corresponding “Materials and Methods” section, where these data should be presented.
Answer:
Amended according to the Reviewer's suggestion.
- The manuscript would benefit from additional editing for typo. For example: page 2, line 78; page 2, line 89; page 3, line 96; page 10, line 331.
Answer:
We apologize for the typing errors. These were corrected throughout the text.
Reviewer 2 Report
I congratulate authors on their efforts to try to clarify an interesting aspect but this reviewer has some major criticism:
Completely different etiology of cirrhosis
Presence of confounding co-morbidities, mainly T2DM and HCC (confirmed by their results)
Complete overlapping of ADMA and NO levels between the two groups
Classification of high and low risk arbitrary based only on MELD and all in all not this high score....see.... End-stage liver disease patients with MELD >40 have higher waitlist mortality compared to Status 1A patients. Hepatol Int 10, 838–846 (2016).Anyway, this classification is the most accepted one.... Status 1 includes patients with acute liver failure/disease with an estimated survival of less than 7 days (highest priority for liver transplantation).Status 2a (MELD score >29) includes patients with end-stage liver disease, severely ill, and potentially hospitalized.Status 2b (MELD score 24–29) includes patients with end-stage liver disease, severely ill, but not requiring hospitalization.Status 3 (MELD score <24) includes patients with liver disease that is too early for cadaveric transplantation but may be a suitable live donor transplantation candidate, Indeed, that said, there are lower cut-offs in the literature showing the arbitrary choice and limits of MELD, see..BMC Gastroenterology;9:72.
Author Response
Completely different etiology of cirrhosis
Answer:
The predominant cause of cirrhosis in our patients was alcohol abuse. The aim of this study was not the investigate different etiologies, which had no impact on the biomarkers of endothelial dysfunction, as emphasized in the Results section, lines 160-161. Rather, we sought to investigate the association of ADMA and NO with estimated CV risk in patients with ESLD, regardless of the etiology.
Presence of confounding co-morbidities, mainly T2DM and HCC (confirmed by their results)
Answer:
There were only 6% of the patients with diabetes in our study cohort, and 28% of patients had HCC on the tops of ESLD (Table 2). Diabetes is a renowned confounding factor regarding both endothelial dysfunction and CV risk. The same goes for the hypertension, which was present in 29% of our ESLD patients (Table 2). These two confounders (diabetes and hypertension) are already included in the SCORE risk assessment, as outlined in the lines 218-220 (Results) and 102 (Methods). Multivariate logistic regression model was designed to evaluate the effects of endothelial and metabolic biomarkers on CV risk, revealing significant, albeit modest effects for the biomarkers of endothelial dysfunction (ADMA and NO) and stronger effect of the fasting plasma glucose, regarded as a continuous variable. However, the contribution of HCC was not so far documented in CV risk in patients with ESLD and requires further investigation, which is emphasized in the Discussion section (lines 297-298) .
Complete overlapping of ADMA and NO levels between the two groups
Answer:
We are not sure about this comment. The results of ADMA and NO differed significantly between MELD-related categories (P=0.013, and <0.001, Table 1) and CV-risk categories (P<0.05 for both biomarkers, Figure 1).
Classification of high and low risk arbitrary based only on MELD and all in all not this high score....see.... End-stage liver disease patients with MELD >40 have higher waitlist mortality compared to Status 1A patients. Hepatol Int 10, 838–846 (2016).Anyway, this classification is the most accepted one.... Status 1 includes patients with acute liver failure/disease with an estimated survival of less than 7 days (highest priority for liver transplantation).Status 2a (MELD score >29) includes patients with end-stage liver disease, severely ill, and potentially hospitalized.Status 2b (MELD score 24–29) includes patients with end-stage liver disease, severely ill, but not requiring hospitalization.Status 3 (MELD score <24) includes patients with liver disease that is too early for cadaveric transplantation but may be a suitable live donor transplantation candidate, Indeed, that said, there are lower cut-offs in the literature showing the arbitrary choice and limits of MELD, see..BMC Gastroenterology;9:72.
Answer:
The comment about the MELD score is valid; however, it refers primarily to transplant communities with a shortage of donors, long waiting times on the transplant lists that result in patients with high MELD scores.
Fortunately, we have the privilege to work in a high volume transplant center in a country with a high donation rate, which results in the highest liver transplantation rate in the world of 30 per million population (https://www.irodat.org/img/database/pdf/Newsletter%20June%202020%20Oct.pdf). Thus, our waiting time on the list is extremely short (in average 21 days), and our patients are "too" sick at the time of transplant, as shown in table 2., where the maximum MELD score was up to 34. MELD score is not an optimal allocation method as specific conditions such as HCC are not reflected in the formula, which involves only laboratory values (bilirubin, creatinine, INR, and sodium). Given that 28% of our patients had hepatocellular carcinoma, their disease's severity is not reflected in the laboratory MELD, and these patients in our center usually have lower MELD scores. For the conditions in which the progression of the disease is expected (such as HCC) and not properly reflected in the lab MELD, the Eurotransplant allocation system grants extra points. These points are called standard exception (SE) (for example, 22 points for HCC at the time of the placement on the list, which automatically increases after several months); and the patients are allocated according to SE MELD. The data presented in Table 2 are the lab. MELD data from our center.
The rationale to select a MELD-related cut-off is explained now in the Methods section (lines 128-130).
Round 2
Reviewer 2 Report
Authors answered my comments
